# Connecting Where You Look With What You Understand: Trajectory-Driven Localized Understanding for Interactive Vision-Language Models

## Abstract

Recent Large Vision Language Models (LVLMs) demonstrate remarkable capabilities in image understanding and natural language generation. However, current approaches focus predominantly on global image understanding, struggling to simulate human visual attention trajectories and explain associations between generated descriptions and specific image regions. To address this challenge, we propose TraceVLM, a unified vision-language model that integrates trajectory-aware spatial understanding within an end-to-end framework. TraceVLM employs a Trajectory-aware Visual Perception (TVP) module for deep bidirectional fusion of visual features and trajectory information. We utilize a geometric simplification algorithm to extract semantic keypoints from raw trajectories and propose a three-stage training pipeline where trajectory information guides description generation and region localization. We further extend TraceVLM to attention trajectory-guided segmentation and video scene understanding tasks, enabling cross-frame trajectory tracking and temporal attention analysis. Based on large vision-language model reasoning capabilities, we construct the Reasoning-based Interactive Localized Narratives (RILN) dataset to enhance logical reasoning and interpretability. Extensive experiments on trajectory-guided captioning, text-guided trajectory prediction, image understanding, and segmentation demonstrate that TraceVLM achieves state-of-the-art performance, establishing a foundation for intuitive human-computer spatial interaction and interpretable visual understanding.

## 1 Introduction

Emerging Large Vision-Language Models (LVLMs) Liu et al. (2023b); Zhu et al. (2023); Dai et al. (2024); Hui et al. (2024) make significant progress in image understanding and natural language generation, and they achieve strong performance on multimodal tasks such as Visual Question Answering (VQA) Antol et al. (2015); Jia et al. (2024) and Image Captioning Lin et al. (2014). However, existing LVLMs still show limitations in spatial attention modeling. As illustrated in the top-left example of Fig. 1, they often focus their attention on the primary regions of an image while neglecting surrounding contextual information, and may even be distracted by irrelevant areas. In contrast, humans naturally guide their visual attention through finger movements or gestures, which helps them understand complex visual content more effectively. Research on human visual attention trajectories also plays an important role in domains such as virtual reality and autonomous driving Xia et al. (2018); Makrigiorgos et al. (2019); Chung et al. (2022).

To address this challenge, researchers have explored incorporating region information into LVLMs for controllable text generation You et al. (2023); Zhang et al. (2024a); Yuan et al. (2024); Ma et al. (2024); Zhang et al. (2024b); Zhao et al. (2024). These methods utilize regional elements such as bounding boxes, masks, points, or coordinates combined with corresponding textual prompts to guide LVLMs in generating localized captions. However, these approaches primarily rely on static discrete localization elements, struggling to model the continuity and temporal characteristics of human visual attention. This limitation reveals several key challenges: (1) existing methods fail

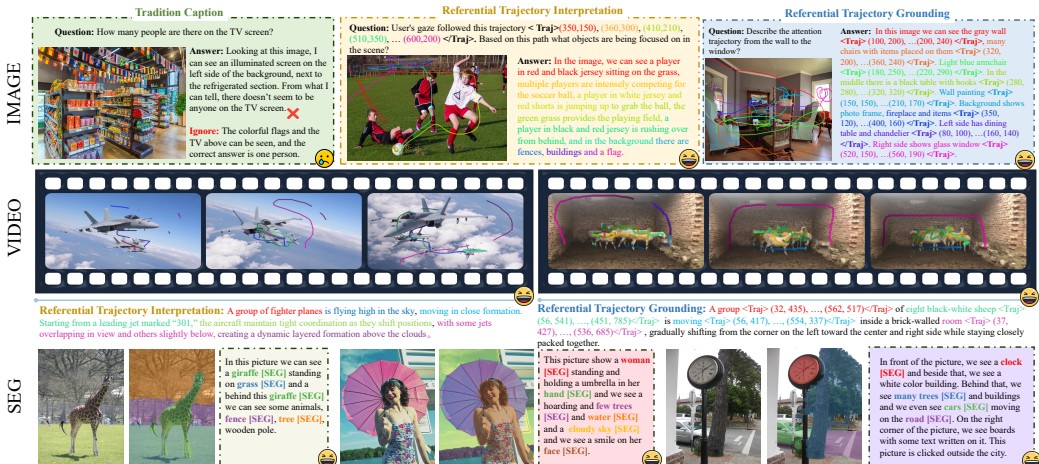

Figure 1. Multi-modal capabilities of TraceVLM across image, video, and segmentation tasks. The model processes traditional captioning, trajectory-guided interpretation and grounding, video sequence analysis, and precise segmentation, demonstrating its versatility in handling diverse visual understanding scenarios with trajectory-based spatial reasoning.

to capture the complex interaction between continuous spatial attention and linguistic expressions; (2) human attention trajectories involve temporal dynamics that require model understanding; (3) existing datasets lack rich trajectory-text alignment training data. This raises a fundamental research question: *how can LVLMs be enhanced to understand and respond to continuous spatial attention patterns?*

In this paper, we present TraceVLM, an end-to-end large vision-language model designed to directly predict and interpret human attention trajectories, treating them as fine-grained and temporally structured records of human attention. TraceVLM models trajectories as dense and expressive signals of human intent and applies Geometric Simplification to extract semantically meaningful keypoints from raw trajectories, effectively reducing redundancy and noise while preserving geometric structures. Furthermore, we design a Trajectory-aware Visual Perception (TVP) module, which captures the sequential patterns of irregular trajectories and deeply integrates them with visual features, enabling precise localization while maintaining global context. Beyond static image tasks such as referential trajectory interpretation and referential trajectory grounding shown in Fig. 1, TraceVLM extends to video understanding by processing multiple frames as input to attention trajectories and to accurate region segmentation by leveraging trajectory guidance.

To ensure the robustness of TraceVLM in understanding trajectory-related multi-dimensional semantic tasks, high-quality training data is essential. Existing datasets, such as Localized Narratives (LN) Pont-Tuset et al. (2020), provide valuable image–text–trajectory alignment but are limited to simple descriptive narratives and lack complex reasoning and instruction-following capabilities. To address this limitation, we introduce the Reasoning-based Interactive Localized Narratives (RILN) dataset, specifically designed to enhance the model logical reasoning and spatial understanding abilities. We develop an advanced data construction pipeline that leverages state-of-the-art VLMs, including GPT-4o Achiam et al. (2023), Qwen2.5VL-72B Bai et al. (2025), and Gemini-2.5 Pro Comanici et al. (2025), to automatically generate 320k high-quality instructional samples, covering diverse tasks such as referential trajectory interpretation, referential trajectory grounding, and interactive trajectory reasoning QA. Based on this dataset, we conduct a comprehensive evaluation of TraceVLM. Experimental results show that TraceVLM effectively exploits trajectory information, achieving state-of-the-art performance on trajectory-conditioned captioning and text-guided trajectory prediction, while also delivering competitive results on Regional Captioning and Referring Localization and Segmentation tasks.

**The main contributions of this work are:**

1. We propose **TraceVLM**, the first end-to-end LVLM that models human attention trajectories for bidirectional trajectory–language understanding.

2. We design a **TVP module** and a **Geometric Simplification** strategy to fuse irregular trajectories with visual features for precise spatial reasoning.

3. We build the **RILN dataset** (320k samples) and demonstrate state-of-the-art performance on trajectory-guided captioning, prediction, and joint generation tasks.

## 2 RELATED WORK

**Vision-Language Models and Regional Understanding.** Traditional image captioning methods use generative models Chen & Lawrence Zitnick (2015); Cornia et al. (2020) with visual attention mechanisms, but these learned attentions often deviate from human perception. Controlled image captioning Xu et al. (2015) addresses this by incorporating bounding boxes or mouse trajectories. Recent large vision-language models like Flamingo Alayrac et al. (2022), BLIP2 Li et al., and LLaVA Liu et al. (2023b) excel at image-level tasks but lack precise spatial localization capabilities for region-level tasks.

**Trajectory-Guided Visual Understanding.** Visual grounding tasks focus on localizing regions based on text queries, with datasets like RefCOCO Yu et al. (2016) and Flickr30K Plummer et al. (2017). Most methods Liu et al. (2017) treat this as a sparse box selection task. Localized Narratives Pont-Tuset et al. (2020) provides dense word-region alignment through simultaneous recording of annotators' voices and mouse traces. Recent extensions include Localized Narratives Video (LNV) Voigtlaender et al. (2023) for video domains and Panoptic Narrative Grounding (PNG) González et al. (2021) for segmentation understanding.

**Human Attention Trajectory Modeling.** This emerging task aligns image descriptions with human attention trajectories collected during annotation. MITR Meng et al. (2021) trains linear layers on frozen features but lacks transferability. PixelLLM Xu et al. establishes pixel-level language-vision correspondence, while Xing et al. (2025) reveals attention mechanism limitations. RegionVLM Lee et al. (2024) pioneers zero-shot regional understanding by converting trajectory points to text tokens, demonstrating strong performance on referring segmentation tasks. Our TraceVLM proposes the first end-to-end framework for trajectory-conditioned captioning, text-guided trajectory prediction, and joint generation tasks, advancing fine-grained visual-text alignment through unified continuous spatial attention modeling.

## 3 METHODOLOGY

As illustrated in Figure 2, TraceVLM is a unified large vision-language model designed to process human attention trajectories bidirectionally for enhanced spatial reasoning and interpretability. Given input image $I \in \mathbb{R}^{H \times W \times 3}$, our primary objective is to establish robust correspondences between visual content, textual descriptions, and human attention patterns through trajectory-aware modeling.

### 3.1 PRELIMINARY

Traditional image captioning methods generate text sequences $S = [w_1, w_2, \ldots, w_n]$ for given images, primarily relying on global image understanding without considering spatial attention mechanisms. While existing large vision-language models perform well on image-level tasks, they remain limited in modeling human visual attention and explaining how generated descriptions correspond to specific image regions.

**Human Attention Trajectory Modeling.** Inspired by human cognitive processes, where eye movements and gestures naturally guide visual attention, we formalize human attention trajectories as continuous temporal point sequences $T = \{p_1, p_2, \ldots, p_N\}$, with each point $p_j = (x_j, y_j, t_j)$ encoding spatial coordinates and temporal information. Unlike static, discrete localization elements, trajectories preserve the continuity and temporal dynamics of human visual exploration, providing richer signals that more faithfully reflect cognitive understanding.

**Trajectory-Text Alignment Mechanism.** To align trajectories with language, we exploit the precise word-level temporal annotations $(w_k, t_k^{\text{start}}, t_k^{\text{end}})$ provided by the Localized Narratives dataset. For each word $w_k$, we extract its corresponding sub-trajectory $T_k$, thereby establishing dense spa-

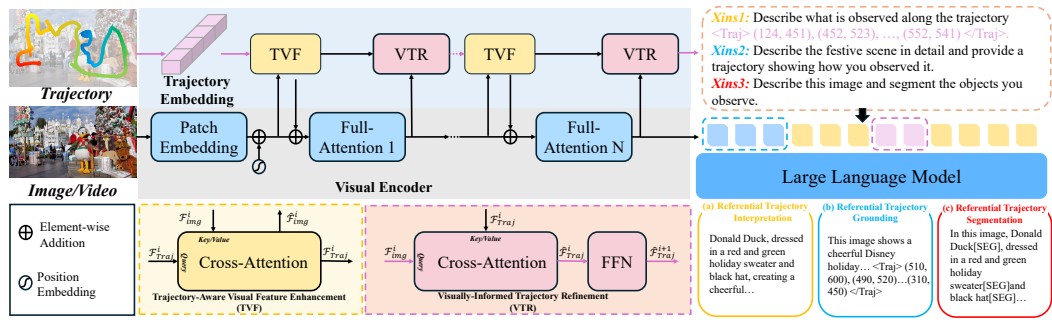

Figure 2. **TraceVLM architecture overview.** The model processes trajectory, image, and text inputs through a unified framework. The TVP module performs bidirectional fusion between visual and trajectory features via cross-attention, enabling trajectory-conditioned captioning, text-guided trajectory prediction tasks.

tial–semantic correspondences that support fine-grained multimodal understanding and bidirectional trajectory–text processing.

**Extensions to Video and Segmentation.** We further extend TraceVLM to video scene understanding by introducing multi-frame visual inputs $I_1, I_2, \ldots, I_T$, where temporal trajectory modeling supports coherent long-sequence video description. In segmentation tasks, we adopt a trajectory-guided strategy by introducing a special [SEG] token and integrating it with a segmentation decoder for prediction. This design allows trajectories to specify interactive target regions, thereby enhancing spatial understanding and supporting fine-grained segmentation of objects of interest.

### 3.2 MODEL ARCHITECTURE

TraceVLM is built upon a unified end-to-end framework that integrates visual encoding, a large language model, trajectory processing, the Trajectory-aware Visual Perception (TVP) module, and a segmentation module to enable efficient modeling and training of human visual attention trajectories.

#### 3.2.1 VISUAL ENCODER

We employ QwenViT as the visual encoder, which effectively processes visual features at arbitrary resolutions. Given input images $I$ and a sequence of video frames $I_1, I_2, \ldots, I_t$ the encoder transforms them into comprehensive visual representations $f_{\text{visual}} \in \mathbb{R}^{N \times D}$, where $N$ denotes the number of visual tokens and $D$ represents the feature dimension. This encoding process captures rich spatial context and temporal information necessary for trajectory-visual alignment

#### 3.2.2 LARGE LANGUAGE MODEL

TraceVLM leverages Qwen2.5-VL-7B, a large language model that unifies the processing of visual features $f_{\text{visual}}$, textual tokens $f_{\text{text}}$ and trajectory representations $f_{\text{Traj}}$ formatted as text. This design supports trajectory-conditioned captioning, text-guided trajectory prediction and trajectory-guided segmentation.

#### 3.2.3 TRAJECTORY PREPROCESSING AND TOKENIZATION

Raw human attention trajectories contain substantial noise and redundancy that obscure true attention patterns. We address this through a two-stage preprocessing approach combining geometric simplification and semantic tokenization.

**Semantic-Guided Geometric Simplification.** Unlike naive uniform sampling which fails to retain critical details, our approach employs a semantic-guided variant of the Douglas-Peucker (DP) algorithm that modulates sampling intensity based on the semantic weight of underlying linguistic content. We first segment the continuous trajectory $T = \{p_1, p_2, \ldots, p_N\}$ into disjoint word-aligned segments $\{S_1, S_2, \ldots, S_M\}$ guided by temporal boundaries of each word $W_i$:

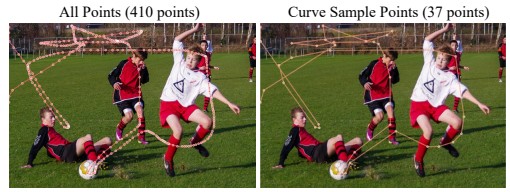

Figure 3. Trajectory simplification: Geometric Simplification algorithm reduces 410 original points to 37 keypoints while preserving spatial structure.

$$S_i = \left\{ p_j \in T \mid t_j \in \left[ t_{\text{start}}^{(i)}, t_{\text{end}}^{(i)} \right] \right\} \tag{1}$$

We employ Qwen2.5-VL-72B model to segment trajectories into semantically meaningful phrases and assign importance weights based on the model's linguistic understanding. Each phrase receives adaptive weighting that reflects its semantic significance in the overall context. For each phrase-aligned segment $S_i$, we apply the Douglas-Peucker algorithm with dynamically calculated local tolerance $\epsilon_i = \frac{\epsilon_{\text{base}}}{w_i}$, where $\epsilon_{\text{base}}$ is set to 5 pixels and $w_i$ represents normalized semantic weight of the phrase (ranging from 0.2 to 1.0, derived from importance scores of 1-5 assigned by Qwen2.5-VL). This ensures semantically important segments preserve more geometric detail critical phrases (importance 5, $w_i$=1.0) maintain maximum detail with $\epsilon_i$=5px, while minimal phrases (importance 1, $w_i$=0.2) allow aggressive simplification with $\epsilon_i$=25px.

For each trajectory segment $T_k = \{(x_i, y_i)\}_{i=s}^e$, the algorithm computes perpendicular distances from intermediate points to the line connecting endpoints:

$$d_i = \frac{|(y_e - y_s)\, x_i - (x_e - x_s)\, y_i + x_e y_s - y_e x_s|}{\sqrt{(y_e - y_s)^2 + (x_e - x_s)^2}} \tag{2}$$

Points with $d_i > \epsilon_i$ are preserved as keypoints. As is show in Fig. 3 this process effectively reduces trajectory density from 410 original points to 37 essential keypoints, achieving 91% compression while maintaining spatial structure and eliminating redundant noise. The final trajectory $T' = \bigcup_{i=1}^M \text{DP}(S_i, \epsilon_i)$ maintains both geometric fidelity and semantic relevance.

**Trajectory Tokenization.** Simplified coordinates are normalized to $[0, 1]$ range and quantized into 1000 discrete bins per dimension, then mapped to discrete tokens compatible with language model vocabularies. We represent coordinate pairs as formatted sequences: $\langle$ traj $\rangle (x_1, y_1), (x_2, y_2), \ldots, (x_n, y_n) \langle$/traj $\rangle$, where each coordinate value is represented as an integer token from 0 to 999, enabling seamless integration with text-based generation.

### 3.2.4 TRAJECTORY-AWARE VISUAL PERCEPTION (TVP) MODULE

The TVP module performs deep bidirectional fusion of visual and trajectory information through iterative refinement. Unlike traditional methods that rely on static annotations, TVP captures temporal dynamics of human visual attention through alternating enhancement and refinement stages.

The module first tokenizes trajectory sequences $T$ and projects them into embedding space, producing trajectory features $f_{\text{Traj}}$. The core fusion process operates through two alternating stages within each block $i$:

**Trajectory-Aware Visual Enhancement.** Visual features $f_{\text{img}}^i$ are enriched with trajectory guidance through cross-attention, where visual features serve as queries and trajectory embeddings as keys and values. The output is scaled by learnable parameter $\gamma^i \in \mathbb{R}^D$ (initialized to zero) for training stability:

$$\hat{f}_{\text{img}}^{i+1} = f_{\text{img}}^i + \gamma^i \cdot \text{Cross Attn}\left(Q = f_{\text{img}}^i, K = f_{\text{Traj}}^i, V = f_{\text{Traj}}^i\right) \tag{3}$$

**Visually-Informed Trajectory Refinement.** Enhanced visual features $\hat{f}_{\text{img}}^i$ refine trajectory representations through a second cross-attention step, followed by feed-forward processing:

$$\hat{f}_{\text{Traj}}^{i+1} = f_{\text{Traj}}^i + \text{CrossAttn}\left(Q = f_{\text{Traj}}^i, K = f_{\text{img}}^{i+1}, V = f_{\text{img}}^{i+1}\right) + \text{FFN}\left(f_{\text{Traj}}^i\right) \tag{4}$$

This bidirectional refinement creates robust multimodal embeddings that effectively integrate spatial attention patterns with visual understanding.

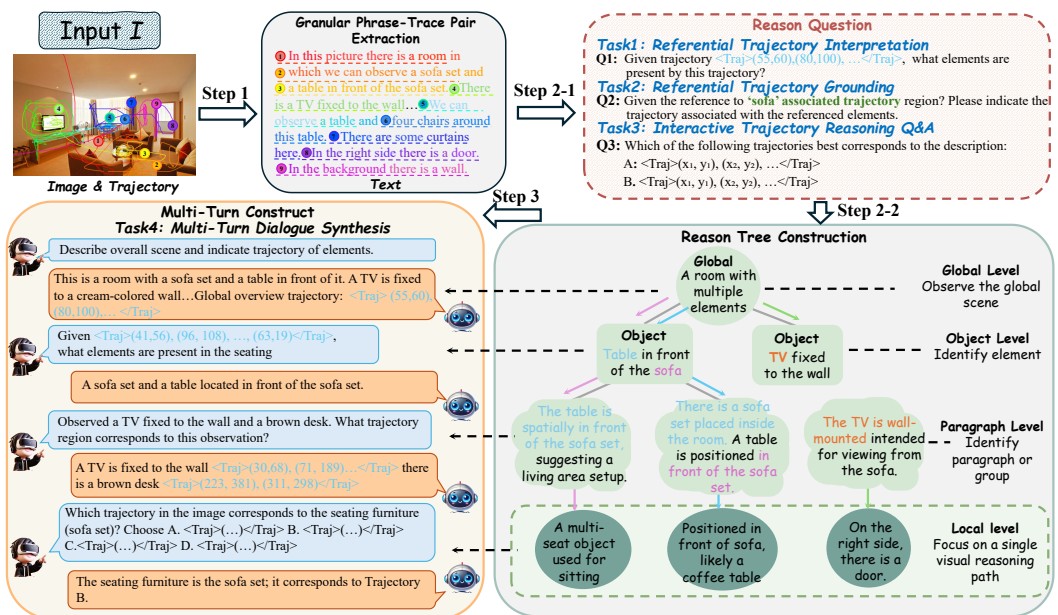

Figure 4. RILN dataset construction pipeline showing the generation of diverse trajectory-based tasks from image-trajectory pairs. The pipeline creates four main task types: referential trajectory interpretation, grounding, interactive reasoning Q&A, and multi-turn dialogue synthesis, with hierarchical reasoning trees spanning from global scene understanding to fine-grained object-level spatial reasoning.

### 3.2.5 SEGMENTATION MODULE

To extend TraceVLM with fine-grained spatial understanding, we introduce a segmentation codebook and a lightweight segmentation decoder, following the efficient design of PixelLM. The segmentation codebook uses 6 learnable embeddings that that encodes semantic and geometric information, which are integrated with visual features $f_{visual}$ and trajectory representations $f_{Traj}$ to provide trajectory-guided spatial priors. When the language model generates a [SEG] token, we extract its corresponding embedding as trajectory-conditioned segmentation tokens $E_{seg}$. The decoder generates pixel-level masks as:

$$\hat{M} = D(f_{visual}, E_{seg}), \tag{5}$$

where $\hat{M}$ denotes the predicted mask and $D$ is a lightweight pixel decoder. The model is trained end-to-end using an overall objective that combines three loss components: an autoregressive cross-entropy loss $\mathcal{L}_{txt}$ for text generation (including segmentation codebook tokens), a DICE loss $\mathcal{L}_{txt}$ for mask overlap, and a target refinement loss $\mathcal{L}_{ref}$ for boundary accuracy. The overall loss is formulated as:

$$\mathcal{L} = \mathcal{L}_{txt} + \lambda_{ref} \mathcal{L}_{ref} + \lambda_{dice} \mathcal{L}_{dice} \tag{6}$$

where $\lambda_{ref}$ and $\lambda_{dice}$ calibrate the contribution of each mask-related loss component. This design enables accurate segmentation without relying on heavy decoders such as SAM and Mask2Former, and naturally generalizes to video segmentation by applying the decoder on multi-frame features $\{f_{visual}^t\}_{t=1}^{T}$ with trajectory guidance across frames.

### 3.3 RILN DATASET CONSTRUCTION

To enhance TraceVLM trajectory-aware reasoning capabilities, we develop the Reasoning-based Interactive Localized Narratives (RILN) dataset through a comprehensive data construction pipeline in Fig 4. While existing datasets like Localized Narratives provide basic trajectory-text alignment, they lack complex reasoning and instruction-following capabilities essential for advanced spatial understanding.

**Dataset Design and Construction Pipeline.** We build RILN based on the COCO Lin et al. (2014), ADE20K Zhou et al. (2017), Flickr30k Plummer et al. (2017), and OpenImage Kuznetsova et al. (2020) portions of the LN dataset, as well as video data from OVIS Qi et al. (2022), UVO Wang et al. (2021), and Oops Epstein et al. (2019) in LNV, using only training splits to ensure evaluation independence. We employ multi-model collaborative generation to avoid single-model bias and enhance reasoning diversity, generating a total of 320k instructional samples through three stages:

**Stage 1: Visual Element and Trajectory Grounding.** We establish precise correspondences between visual elements and trajectory segments through geometric simplification for keypoint extraction, followed by word-point alignment and phrase-trajectory segmentation using Qwen2.5VL-72B.

**Stage 2: Hierarchical Reasoning Task Generation.** We define three core reasoning categories: referential trajectory interpretation, referential trajectory grounding, and interactive trajectory reasoning QA. GPT-4o generates question prompts leveraging its strength in logical coherence, while Gemini-2.5 Pro constructs structured reasoning trees and produces corresponding answers with diverse expression styles. The reasoning tasks are organized across four cognitive levels (Global, Object, Paragraph, Local) as a reference framework rather than rigid templates. We randomly vary expression styles, reasoning structures, and detail levels to enhance diversity and adapt to different scene complexities, ensuring a progressive transition from holistic scene understanding to fine-grained spatial analysis.

**Stage 3: Multi-Turn Dialogue Synthesis.** We synthesize 3-5 coherent dialogue sequences per image with 4-8 turns each, integrating hierarchical reasoning tasks into natural conversational flows. Final turns require synthesis of conversational context with trajectory information for comprehensive spatial understanding.

**Quality Assurance.** We employ multi-level verification, including automatic consistency checking and human expert annotation on 2000 samples, ensuring n-gram overlap (n=5) between RILN and evaluation annotations stays below 5% to prevent textual leakage. Models trained on RILN achieve 23% improvement in spatial reasoning accuracy compared to baseline LN training, validating our construction methodology effectiveness.

### 3.4 TRAINING STRATEGY

We adopt a three-stage curriculum learning approach that progressively builds TraceVLM capabilities from basic trajectory-visual alignment to complex reasoning and dialogue understanding.

**Stage 1: Trajectory-Aware Pretraining.** We establish robust alignment between trajectory features, visual features, and language embeddings using large-scale trajectory-text-image data. Only TVP modules and trajectory embedding layers are trained, focusing on foundational cross-modal representations for trajectory-guided captioning, caption-guided trajectory prediction, and joint generation tasks.

**Stage 1.5: End-to-End Joint Training.** We unfreeze all module parameters and perform full joint training on the visual encoder, large language model, TVP module, and segmentation decoder to optimize the collaborative mechanisms between components and enhance the overall multimodal fusion capability of the system.

**Stage 2: Instruction Fine-tuning.** We employ supervised fine-tuning on RILN dataset to enable complex reasoning and conversational capabilities. Joint optimization of TVP modules, trajectory embeddings, and language decoder adapts the model to instruction-following scenarios while preserving robust multimodal representations from pretraining stages.

## 4 EXPERIMENT

### 4.1 TRAINING DETAILS.

We build TraceVLM on the Qwen2.5-VL-7B model through a three-stage training approach. In Stage 1, we freeze the visual encoder and language model while fine-tuning only the TVP module for 1 epoch, setting the batch size to 256 and learning rate to 4e-5. In Stage 1.5, we train all parameters for 3 epochs to strengthen foundational capabilities, with batch size 128 and learning

Table 1. Quantitative results for trajectory-aware vision-language tasks on the COCO test set of Localized Narratives dataset. Controlled Caption Generation refers to generating descriptions given trajectory input, while Controlled Trajectory Generation refers to predicting trajectories given text input. Note: smaller values of LBM are better. Best results are in bold and second-best results are underlined.

| Method | Contrlled Caption Generation | | | | | | Controlled Trace Generation | |
|---|---|---|---|---|---|---|---|---|
| | BLUE-1↑ | BLUE-4↑ | METEOR↑ | ROUGE_L↑ | CIDEr↑ | SPICE↑ | LBM(k=0)↓ | LBM(k=1)↓ |
| LN Pont-Tuset et al. (2020) | 0.522 | 0.246 | - | 0.483 | 1.065 | 0.365 | - | - |
| MITR Meng et al. (2021) | 0.607 | 0.292 | 0.263 | 0.487 | 1.485 | 0.317 | 0.163 | 0.154 |
| PIxelLLM Xu et al. | - | - | - | - | - | - | 0.153 | - |
| LLaVA 1.5-13B Liu et al. (2023b) | 0.590 | 0.280 | 0.250 | 0.480 | 1.450 | 0.310 | - | - |
| Ferret-13B You et al. (2023) | 0.600 | 0.283 | 0.255 | 0.482 | 1.470 | 0.315 | 0.160 | 0.170 |
| Qwen2.5 VL-7B Hui et al. (2024) | 0.630 | 0.295 | 0.260 | 0.486 | 1.500 | 0.320 | 0.147 | 0.168 |
| **TraceVLM-7B** | **0.665** | **0.328** | **0.276** | **0.492** | **1.530** | **0.328** | **0.117** | **0.121** |

Table 2. Performance comparison on trajectory-aware regional captioning across multiple benchmarks. TraceVLM demonstrates competitive performance on Visual Genome (VG), RefCOCOg, and Ref-L4 datasets, as well as benchmarks including Ferret-Bench and MDVP-Bench for spatial understanding evaluation.

| Model | VG | | RefCOCOg | | Ref-L4 | | | Ferret Bench | MDVP Bench |
|---|---|---|---|---|---|---|---|---|---|
| | METEOR | CIDEr | METEOR | CIDEr | ROUGE-L | METEOR | CIDEr | Refer. Desc. | Avg. |
| PixelLLM | 19.9 | 148.9 | 14.3 | 82.3 | - | - | - | - | - |
| GLaMM-7B Rasheed et al. (2024) | 17.0 | 127.0 | 15.7 | 104.0 | 23.8 | 10.1 | 51.1 | - | - |
| Osprey-7B Yuan et al. (2024) | - | - | 16.6 | 108.3 | - | - | - | 72.2 | 44.3 |
| Ferret-7B You et al. (2023) | - | - | - | - | 22.3 | 10.7 | 39.7 | 68.7 | 47.6 |
| VP-LLaVA-8B Lin et al. (2024) | - | - | 22.4 | 153.6 | - | - | - | 75.2 | 70.6 |
| VP-SPHINX-7B Lin et al. (2024) | 20.1 | 139.5 | 21.0 | 138.7 | 22.5 | 10.5 | 32.2 | 73.1 | 71.5 |
| VP-SPHINX-13B Lin et al. (2024) | 20.6 | 141.8 | 23.9 | 162.5 | 22.6 | 10.7 | 32.4 | 77.4 | **74.3** |
| Omni-RGPT-7B Heo et al. (2025) | 17.0 | 139.3 | 17.0 | 109.7 | - | - | - | - | - |
| RegionGPT-7B Guo et al. (2024) | 17.0 | 145.6 | 16.9 | 109.9 | 25.3 | 12.2 | 42.0 | - | - |
| DAM-8B Lian et al. (2025) | - | - | - | - | 37.1 | 19.4 | 70.0 | - | - |
| PAM-3B Lin et al. (2025) | 20.8 | 142.3 | 26.9 | 143.1 | 31.3 | 17.2 | 59.7 | **77.5** | 72.2 |
| **TraceVLM-7B** | **21.5** | **163.2** | **28.8** | **168.2** | 36.5 | 20.3 | 69.7 | 76.7 | 73.1 |

rate 4e-5. In Stage 2, we freeze the visual encoder and TVP while fine-tuning only the language model using instruction data for 1 epoch, reducing the learning rate to 2e-5 and maintaining batch size at 128. We trained the model on 16 A800 GPUs for approximately 3 days total across all stages.

## 4.2 COMPARISON CONTROLLED CAPTION AND TRACE GENERATION

In the table 1, we evaluate two trajectory-aware tasks: Controlled Caption Generation (generating descriptions from images and trajectories) and Controlled Trajectory Generation (predicting trajectories from text descriptions).

**Controlled Caption Generation.** Given an image and trajectory, the model generates corresponding captions evaluated using BLEU, METEOR, ROUGE-L, CIDEr, and SPICE metrics. We compare against LVLMs using task-specific prompts and PixelLLM as baseline. TraceVLM consistently outperforms all methods, demonstrating superior trajectory-conditioned description quality.

**Controlled Trajectory Generation.** The model predicts continuous point sequences given input captions, evaluated using LBM scores (k=0, k=1) for spatial alignment accuracy. We compare against MITR (bounding box sequences) and adapted LVLMs with prompted trajectory generation. TraceVLM outperforms baseline methods in LBM scores, validating its trajectory prediction effectiveness.

## 4.3 COMPARISON REGIONAL CAPTION

As is shown in table 2, we evaluate TraceVLM on regional captioning benchmarks (RefCOCOg, Visual Genome, Ref-L4) and spatial understanding benchmarks (Ferret-Bench, MDVP-Bench) using ROUGE-L, METEOR, and CIDEr metrics. TraceVLM-7B achieves state-of-the-art performance with METEOR scores of 21.5 on VG and 28.8 on RefCOCOg, and strong results on Ref-L4 (ROUGE-L: 36.5, METEOR: 20.3). The model also delivers competitive performance on Ferret-Bench (76.7) and MDVP-Bench (73.1), TraceVLM achieves near-SOTA performance with comparable model size. Specifically, VP-SPHINX-7B, the closest 7B competitor, achieves 73.1 and 71.5 respectively, while higher scores from VP-SPHINX-13B (77.4, 74.3) and PAM-3B (77.5, 72.2) come from models with either nearly 2× parameters or extensive specialized visual processing. Notably, TraceVLM achieves these results using only a lightweight 2-layer segmentation decoder, demonstrating effective trajectory-aware visual perception without heavy architectural overhead.

| Models | | RefCOCO | | | RefCOCO+ | | | RefCOCOg | |
|---|---|---|---|---|---|---|---|---|---|
| | | val | testA | testB | val | testA | testB | val | test |
| | | bounding box P @ 0.5 | | | | | | | |
| VisionLLM | Wang et al. (2023) | 86.7 | - | - | - | - | - | - | - |
| Shikra-7B | Chen et al. (2023) | 87.0 | 90.6 | 80.2 | 81.6 | 87.4 | 71.1 | 82.3 | 82.2 |
| Ferret-7B | | 87.5 | 91.4 | 82.5 | 80.8 | 87.4 | 73.1 | 83.9 | 84.8 |
| PixelLLM-7B | | 89.8 | 92.2 | 86.4 | 83.2 | 87.0 | 78.9 | 84.6 | 86.0 |
| Qwen2.5-VL-7B | | 90.0 | 92.5 | 85.4 | 84.2 | 89.1 | 76.9 | 87.2 | 87.2 |
| TraceVLM-7B | | 90.4 | 93.1 | 87.8 | 84.6 | 88.2 | 80.5 | 86.9 | 88.3 |
| | | segmentation mask cIoU | | | | | | | |
| RegionVLM-4B* | Lee et al. (2024) | 38.7 | 39.4 | 37.6 | 31.5 | 34.0 | 30.2 | 33.9 | - |
| LISA-7B | Lai et al. (2024) | 74.9 | 79.1 | 72.3 | 65.1 | 70.8 | 58.1 | 67.9 | 70.6 |
| PixelLLM-7B | | 76.9 | 78.5 | 74.4 | 69.2 | 72.1 | 64.5 | 70.7 | 72.4 |
| PixelLM-7B | Ren et al. (2024) | 73.0 | 76.5 | 68.2 | 66.3 | 71.7 | 58.3 | 69.3 | 70.5 |
| PSALM-3B | Zhang et al. (2024b) | 83.6 | 84.7 | 81.6 | 72.9 | 75.5 | 70.1 | 73.8 | 74.4 |
| HyperSeg-1.5B | Nirkin et al. (2021) | 84.8 | 85.7 | 83.4 | 79.0 | 83.5 | 75.2 | 79.4 | 78.9 |
| TraceVLM-7B | | 83.4 | 86.8 | 82.4 | 80.1 | 84.2 | 76.8 | 77.6 | 80.1 |

Table 3. Performance comparison on referring localization and segmentation across RefCOCO datasets. TraceVLM achieves state-of-the-art performance on both bounding box localization (P@0.5) and segmentation (cIoU) tasks. * indicates zero-shot results without task-specific fine-tuning.

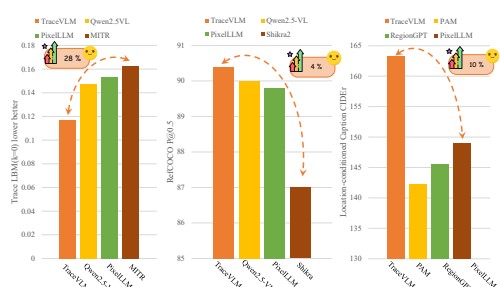

Figure 5. Performance analysis visualization comparing trajectory-aware methods across different evaluation metrics and datasets.

Table 4. TVP module ablation experiment results.

| Method | BLEU-4↑ | METEOR↑ | LBM↓ | |
|---|---|---|---|---|
| | | | (k=0) | (k=1) |
| Baseline | 0.250 | 0.195 | - | - |
| +Text | 0.265 | 0.205 | 0.135 | 0.125 |
| +Vision Encoder | 0.280 | 0.210 | 0.120 | 0.110 |
| +TVP (Ours) | **0.328** | **0.276** | **0.117** | **0.121** |

Table 5. Task train order ablation experiment.

| Method | BLEU-4↑ | METEOR↑ | LBM↓ | |
|---|---|---|---|---|
| | | | (k=0) | (k=1) |
| Baseline | 0.250 | 0.195 | - | - |
| +Random Order | 0.310 | 0.225 | 0.163 | 0.154 |
| +Fixed Order | **0.328** | **0.276** | **0.117** | **0.121** |

Table 6. RILN training data ablation experiment.

| Method | BLEU-4↑ | METEOR↑ | LBM↓ | |
|---|---|---|---|---|
| | | | (k=0) | (k=1) |
| Baseline | 0.250 | 0.195 | - | - |
| +LN | 0.267 | 0.224 | 0.143 | 0.148 |
| +RILN | **0.328** | **0.276** | **0.117** | **0.121** |
| Improvement | +23.0% | +23.2% | -18.2% | -18.2% |

## 4.4 COMPARISON REFERRING LOCALIZATION AND SEGMENTATION

As shown in the table. 3, we evaluate TraceVLM on RefCOCO, RefCOCO+, and RefCOCOg using bounding box P@0.5 and segmentation cIoU metrics. TraceVLM achieves state-of-the-art performance across most splits, with bounding box localization scores of 90.4 P@0.5 on RefCOCO val, 84.6 on RefCOCO+ val, and 86.9 on RefCOCOg val. For segmentation, TraceVLM demonstrates competitive performance with 83.4 cIoU on RefCOCO val, 80.1 on RefCOCO+ val, and 77.6 on RefCOCOg val, achieving strong results using only a lightweight 2-layer segmentation decoder. Notably, models with higher segmentation scores, such as HyperSeg, rely on multi-scale visual encoders and the significantly heavier Mask2Former decoder, demonstrating the efficiency advantage of our trajectory-guided approach. These results demonstrate that trajectory guidance effectively enhances both localization and segmentation capabilities with minimal architectural overhead. The performance visualization in Fig. 5 further illustrates TraceVLM consistent advantages across different metrics and datasets.

## 4.5 QUALITATIVE ANALYSIS AND VISUALIZATION

Figure 6 shows trajectory-guided attention heatmaps across diverse scenarios. The three-column visualizations demonstrate that compared to baseline Qwen2.5-VL-7B (column 2), TraceVLM (column 3) accurately focuses on salient objects mentioned in descriptions, producing broader, semantically aligned attention patterns that better capture human visual attention mechanisms.

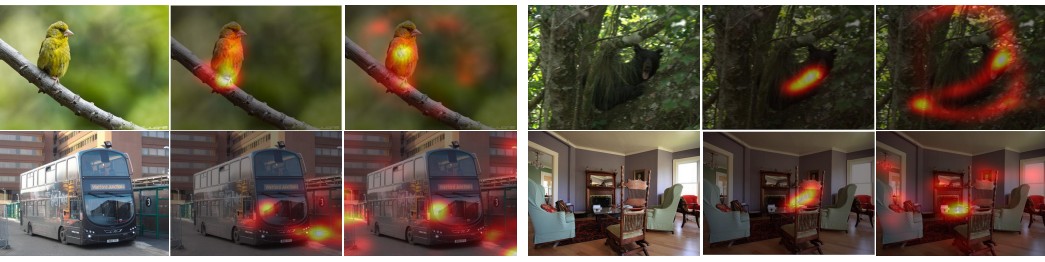

Figure 6. Qualitative trajectory visualization results showing original images (left), attention heatmaps (center), and combined visualizations (right). TraceVLM generates focused heatmaps that align with salient objects mentioned in corresponding descriptions.

4.6 ABLATION STUDY

**TVP Module Ablation.** We compare trajectory representation strategies against the TraceVLM pretraining baseline: text encoding (+Text), visual encoder (+Vision Encoder), and our TVP module. Table 4 shows our TVP module achieves the best performance by effectively fusing trajectory and visual features through cross-attention.

**Task Randomization Strategy Ablation.** We compare our fixed task order training with randomized task combination approach. The randomized strategy dynamically alternates among trajectory-guided caption generation, caption-guided trajectory prediction, and joint generation tasks. Table 5 shows fixed order training outperforms randomized training with higher BLEU-4 (0.328 vs. 0.310), METEOR (0.276 vs. 0.225), and better LBM scores, demonstrating the effectiveness of structured curriculum learning.

**RILN Dataset Ablation.** As shown in Table 6, models trained on RILN significantly outperform both baseline and LN-trained models across all metrics. RILN training achieves 23.0% improvement in BLEU-4 and 23.2% improvement in METEOR scores compared to LN training, while reducing trajectory prediction errors by 18.2% for both LBM(k=0) and LBM(k=1), validating the effectiveness of our dataset construction methodology.

## 5 CONCLUSION

We introduced TraceVLM, a trajectory-aware large vision-language model that processes human attention trajectories bidirectionally to enhance spatial reasoning and interpretability. Through our Trajectory-aware Visual Perception (TVP) module and the Reasoning-based Interactive Localized Narratives (RILN) dataset with 320k instructional samples, TraceVLM addresses fundamental limitations in existing vision-language models by capturing the continuity and temporal dynamics of human visual attention patterns. Comprehensive evaluation demonstrates state-of-the-art performance across trajectory-guided captioning, text-guided trajectory prediction, referring localization, and segmentation tasks, with 23% improvement in spatial reasoning accuracy and effective generation of human-like attention patterns. TraceVLM establishes a foundation for intuitive human-computer spatial interaction and interpretable visual understanding, with future work exploring integration with additional modalities and real-time optimization for interactive applications that better align with human cognitive processes.

**Reproducibility Statement.** The TraceVLM execution source code has been included in the supplementary materials. All execution configurations, relevant parameters, and trajectory sampling implementations are provided within the associated scripts and data projects, facilitating reproducibility of the results.

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

## A APPENDIX

### A.1 THE USE OF LARGE LANGUAGE MODELS(LLMS)

We acknowledge the use of large language models (LLMs) as writing assistants only for grammatical and style. LLMs are not employed in the core research methodology, experimental design, data analysis, or generation of research findings presented in this paper. All textual content has been rigorously reviewed and verified by the authors to ensure accuracy and authenticity of the research contributions.

#### A.1.1 OVERVIEW OF TRACEVLM

TraceVLM is a comprehensive vision-language model that processes single images or multi-frame videos as input, as illustrated in Figure 7. Based on various input modalities including queries, captions, clicks, and trajectories, TraceVLM performs multiple fundamental tasks such as trajectory-guided caption generation, trajectory prediction, and trajectory-guided segmentation. Additionally, the model supports advanced capabilities including referential trajectory interpretation, referential trajectory grounding, interactive trajectory reasoning through question-answering, and multi-turn dialogue interactions.

### A.2 PROGRESSIVE DATASET STRUCTURING

Each stage in TraceVLM adopts dedicated datasets aligned with its training objectives, rather than relying on a uniform corpus across all phases. The dataset distribution and task assignments are summarized in Table 7 and visualized in Figure 8.

**Stage 1: Trajectory-Aware Pretraining.** This stage focuses on establishing robust alignment between trajectory features, visual features, and language embeddings. We utilize 252K trajectory-text-image samples from COCO Localized Narratives. This dataset provides foundational cross-modal representations for trajectory-guided captioning, caption-guided trajectory prediction, and joint generation tasks. Only TVP modules and trajectory embedding layers are trained during this phase.

**Stage 1.5: End-to-End Joint Training.** To enhance multimodal fusion capabilities, we unfreeze all module parameters and perform comprehensive joint training on 1.33M samples across multiple tasks. We leverage 885K samples from Localized Narratives spanning COCO, ADE20K, Flickr30K,

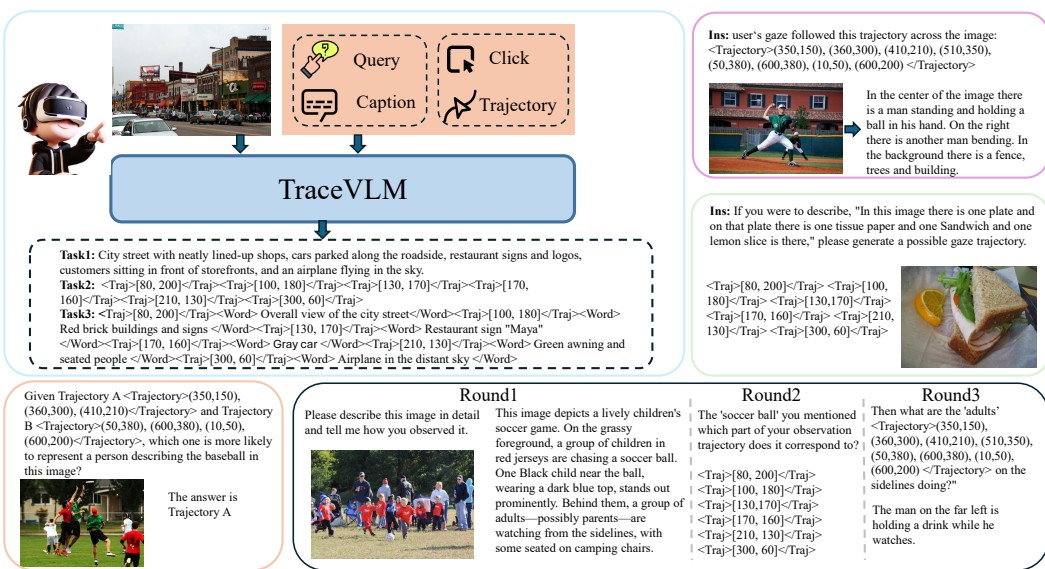

Figure 7. Overview of the TraceVLM framework. The model takes images or multi-frame videos as input and supports multiple interaction modalities (query, caption, click, trajectory) to perform various tasks including trajectory-guided caption generation, trajectory prediction, trajectory-guided segmentation, referential trajectory interpretation, referential trajectory grounding, and interactive trajectory reasoning through QA and multi-turn dialogue.

Table 7. Dataset distribution across training stages in TraceVLM

| Stage | Number | Task | Source |
|---|---|---|---|
| Stage 1: Trajectory-Aware Pretraining | 252K | Localized Narratives | COCO |
| Stage 1.5: End-to-End Joint Training | 885K | Localized Narratives | COCO, ADE20K Flickr30k, OpenImage |
| | 50K | Localized Narrative Video | OVIS, UVO Oops |
| | 118K | Panoptic Narrative Grounding | COCO |
| | 40K | Video Narrative Grounding | OVIS, UVO |
| | 10K | Video Question Answering | Oops |
| | 232K | REC/RES | RefCOCO Series gRefCOCO Liu et al. (2023a) |
| Stage 2: Instruction Fine-turning | 320K | Reasoning-based Interactive | RILN |
| | 480K | Reasoning Segmentation | ReasonSeg Lai et al. (2024) Lisa++ Inst. Seg. & CoT Yang et al. (2023) ReVOS Yan et al. (2024) |
| | 500K | Interactive | COCO-Interactive Zhang et al. (2024b) |

and OpenImage for spatial-textual alignment. Video understanding is supported by 50K samples from Localized Narrative Video across OVIS, UVO, and Oops datasets, enabling video narrative grounding (40K from OVIS and UVO) and video question answering (10K from Oops). Additionally, 118K samples from COCO provide Panoptic Narrative Grounding for segmentation tasks, while 232K samples from RefCOCO Series and gRefCOCO support referring expression comprehension and segmentation (REC/RES). This stage jointly optimizes the visual encoder, large language model, TVP module, and segmentation decoder.

**Stage 2: Instruction Fine-tuning.** The final stage fine-tunes the model for complex reasoning and conversational capabilities using 1.3M samples. We employ 320K samples from our constructed RILN (Reasoning-based Interactive Localized Narratives) dataset to enable instruction-following scenarios while preserving robust multimodal representations. For advanced reasoning segmentation, we incorporate 480K samples from ReasonSeg, Lisa++ Inst. Seg. & CoT, and ReVOS datasets. Additionally, 500K samples from COCO-Interactive provide interactive region-level editing capabilities. This stage performs joint optimization of TVP modules, trajectory embeddings, and language decoder to adapt the model to instruction-following scenarios.

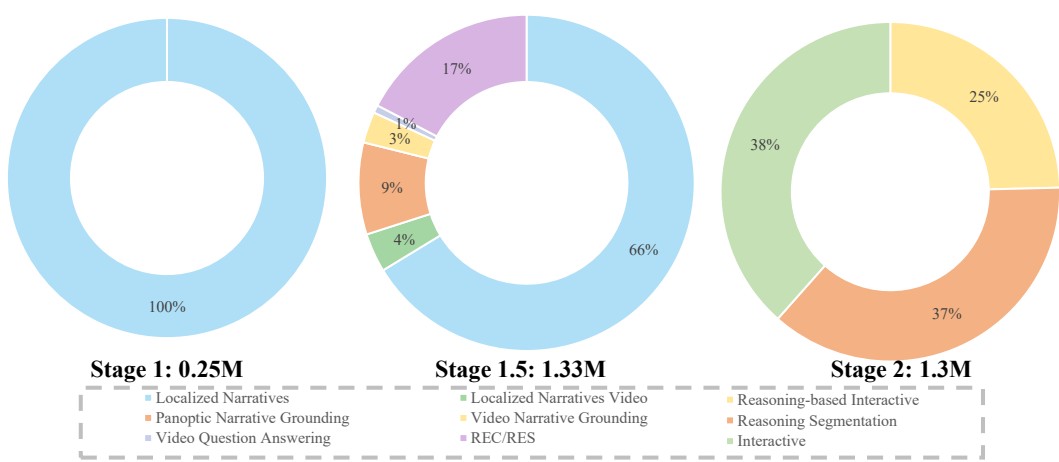

Figure 8. Dataset distribution across the three training stages of TraceVLM. Stage 1 (0.25M samples) focuses solely on trajectory-aware pretraining with COCO Localized Narratives. Stage 1.5 (1.33M samples) incorporates diverse multimodal tasks including localized narratives (66%), video understanding (4%), panoptic grounding (9%), video narrative grounding (3%), video QA (1%), and REC/RES (17%). Stage 2 (1.3M samples) emphasizes instruction fine-tuning with reasoning-based interactive tasks (25%), reasoning segmentation (37%), and interactive editing (38%).

Table 8. Performance comparison on video understanding benchmarks. TraceVLM demonstrates competitive performance across HC-STVG and VideoRefer-Bench-D datasets.

| Model | HC-STVG Tang et al. (2021) | | VideoRefer-Bench-D Yuan et al. (2025) | | | | |
|---|---|---|---|---|---|---|---|
| | METEOR | CIDEr | SC | AD | TD | HD | Avg. |
| Elysium-7B Wang et al. (2024) | – | – | 2.35 | 0.30 | 0.02 | 3.59 | 1.57 |
| Merlin-7B Yu et al. (2024) | 11.3 | 10.5 | – | – | – | – | – |
| Artemis-7B Qiu et al. (2024) | 18.0 | 53.2 | 3.42 | 1.34 | 1.39 | 2.90 | 2.26 |
| VideoRefer-7B Yuan et al. (2025) | 18.7 | 68.6 | 4.44 | 3.27 | 3.10 | 3.04 | 3.46 |
| DAM-8B | 21.0 | 91.0 | 4.69 | 3.61 | **3.34** | 3.09 | 3.68 |
| PAM-3B | 23.3 | 70.3 | 3.92 | 2.84 | 2.88 | 2.94 | 3.14 |
| TraceVLM-7B | **24.1** | **92.6** | **4.73** | **3.89** | 3.11 | **3.28** | **3.83** |

## A.3 VIDEO UNDERSTANDING EVALUATION

"We evaluate TraceVLM on video understanding tasks using HC-STVG for spatio-temporal video grounding and VideoRefer-Bench-D for comprehensive video referring capabilities. As shown in Table 8, TraceVLM achieves state-of-the-art performance across most metrics, demonstrating exceptional video understanding capabilities. On HC-STVG, TraceVLM attains the highest METEOR score of 24.1 and CIDEr score of 92.6, surpassing previous methods in video caption generation quality. For VideoRefer-Bench-D evaluation, TraceVLM delivers the best average performance of 3.83, with top results in Spatial Comprehension (4.73), Action Detection (3.89), and Human Detection (3.28) tasks. These results validate TraceVLM's superior ability to process temporal trajectory information across video sequences, extending its capabilities from static image understanding to dynamic video content analysis with remarkable effectiveness.

