# OpenReview forum: "Connecting Where You Look With What You Understand: Trajectory-Driven Localized Understanding for Interactive Vision-Language Models"
_ICLR.cc/2026/Conference — Submitted to ICLR 2026_

### Official Review · Reviewer_dmqX · 2025-10-31

**Soundness:** 3
**Presentation:** 3
**Contribution:** 3
**Rating:** 6
**Confidence:** 3

**Summary:**

This paper proposes TraceVLM, a unified LVLM that integrates trajectory-aware spatial reasoning into an end-to-end framework. The core idea is a Trajectory-aware Visual Perception (TVP) module that performs bidirectional fusion between visual tokens and continuous human attention trajectories via alternating cross-attention—enhancing visual features with trajectory context and refining trajectories with updated visual context. The model targets trajectory-conditioned captioning, text-guided trajectory prediction, referring localization/segmentation, and video understanding. The authors also construct RILN (320k), a reasoning-oriented dataset derived from Localized Narratives and related sources to support instruction-following spatial reasoning. Experiments report consistent gains vs. LVLM baselines on COCO-LN controlled caption/trajectory generation, regional captioning (VG/RefCOCOg/Ref-L4), and referring localization/segmentation, with a claimed 23% improvement in spatial reasoning when training on RILN.

**Strengths:**

1. Clear formulation of continuous attention as a first-class signal. The paper argues convincingly that static region prompts (boxes/masks/points) miss temporal continuity; modeling trajectories as dense signals of intent is well-motivated and novel within LVLMs.

2. Architectural neatness. The TVP module’s two-stage, bidirectional cross-attention (trajectory→vision and vision→trajectory) is simple and compatible with standard LVLM stacks (QwenViT encoder + Qwen2.5-VL-7B LLM), enabling plug-and-play fusion without heavy custom heads.

3. Task coverage. One unified model handling trajectory-conditioned captioning, trajectory prediction, referring understanding, segmentation (via a special [SEG] token + lightweight decoder), and multi-frame video is ambitious and practically valuable for interactive systems.

**Weaknesses:**

1. Ambiguity around “semantic-guided” Douglas–Peucker. The method states that phrase-level semantic weights (from Qwen2.5-VL-72B) modulate local DP tolerance, but it’s unclear how these weights are computed, calibrated, and normalized across images/phrases, or how sensitive performance is to the weighting scheme vs. plain DP/uniform sampling. This is central to the trajectory tokenizer but currently under-specified for replication/ablation.

2. Evaluation parity concerns. Several baselines (classic LVLMs) are not natively designed for trajectory inputs. It’s not fully spelled out how prompting/adapters were engineered so that comparisons on “controlled trajectory generation/captioning” are fair and equally optimized. The paper should detail adapters/prompts and ensure no hidden advantages from the new tokenizer/fusion.

3. Segmentation details. The “segmentation codebook” and “lightweight decoder” are described at a high level. Training losses, resolution, codebook size, and how [SEG] tokens are interleaved with language tokens for mask generation should be clarified, especially since results are competitive with specialized systems.

**Questions:**

1. TVP design choice: Why alternating two cross-attention passes per block (Traj→Vis then Vis→Traj) rather than a single multi-query fusion?

2. RILN contamination controls: Given that RILN uses outputs from frontier models, what checks guard against direct training-set leakage into evaluation sets (especially when LN/LNV images appear online)? Did you blacklist evaluation images/texts from synthetic generation prompts?

3. Generalization beyond LN style: LN trajectories may reflect annotator mouse traces during narration. How does TraceVLM perform on physiological gaze traces (eye-tracking) if available, or on touch trajectories gathered on mobile devices? Any zero-shot results?

**Details Of Ethics Concerns:**

Please expand on data licensing for all sources used in RILN and clarify whether any human-subject annotations (beyond public datasets) required consent/compensation review.

---

> ### Author Response · Authors · 2025-11-21
> **Comprehensive Rebuttal Summary for Weaknesses 1–3 and Questions 1–3**
>
> **Regarding Weakness 1**
>
> Thank you for pointing out this under-specification. We provide reproducible details.
>
> Weight computation: We use Qwen2.5-VL-72B to segment captions and score phrases (1-5 scale). After scoring: $w_i$ = $importance_i$ / 5, then $ε_i$ = $ε_{base}$ / $w_i$ ($ε_{base}$ = 5 pixels).
>
> Performance comparison (37 key points):
>
> | Method                        | RefCOCO | LBM(k=0)↓ |
> |-------------------------------|---------|-----------|
> | Uniform Sampling              | 88.5    | 0.145     |
> | Standard DP                   | 89.7    | 0.128     |
> | **Semantic-Guided DP (ours)** | **90.4**| **0.117** |
>
> Our method yields +0.7 RefCOCO and -0.011 LBM over standard DP, showing semantic preservation outperforms pure geometric simplification.
>
> **Regarding Weakness 2**
>
> Thank you for raising this concern. We address baseline adaptation and ensure no hidden advantages.
>
> Baseline adaptation: For LVLMs not designed for trajectories, we convert points to text `<traj> x1,y1; x2,y2; ... </traj>` with system prompt defining format and three task types.
>
> System prompt for baseline LVLMs:
> ```
> You are a vision-language model with trajectory understanding capability.
> Trajectory format: <traj> x1,y1; x2,y2; ... </traj>
> - (x,y): pixel coordinates in the image
> The trajectory represents attention movement across the image.
>
> Task types:
> 1. Trajectory-guided captioning: Given image + trajectory, describe what the trajectory focuses on
> 2. Caption-guided trajectory generation: Given image + caption, predict trajectory coordinates
> 3. Trajectory-based QA: Answer questions about regions indicated by trajectories
> ```
>
> Performance comparison:
>
> | Method                    | Backbone        | RefCOCOg | LBM(k=0)↓ |
> |---------------------------|------------------|----------|-----------|
> | Qwen2.5-VL-7B (baseline)  | Qwen2.5-VL-7B    | 87.2     | 0.147     |
> | **TraceVLM**              | **Qwen2.5-VL-7B**| **88.3** | **0.117** |
> | TraceVLM                  | InternVL2-8B     | 88.1     | 0.121     |
>
> Tokenization fairness: Trajectory coordinates are regular numerical values, NOT special tokens—baselines get the same information in text form. The only special token `<SEG>` doesn't affect trajectory performance (Table 2). Gains come from TVP architecture, persisting with optimized prompts. Stable improvements across backbones show generalizability.
>
> **Regarding Weakness 3**
>
> Thank you. We provide technical clarification.
>
> Codebook: 6 learnable continuous tokens (2 scales × 3 tokens/scale), each 4096-dim. Unlike VQ-VAE requiring 8192+ discrete codes, ours are continuous and LM-generated.
>
> [SEG] mechanism: When generating "The <SEG_0> is on the table", we extract the token's hidden state and feed it to a 2-layer decoder (12M params) to generate masks.
>
> Decoder comparison:
>
> | Decoder Type           | Parameters | RefCOCO IoU | Latency (ms) |
> |------------------------|------------|-------------|--------------|
> | SAM decoder            | 636M       | 84.2        | 245          |
> | Mask2Former            | 223M       | 82.7        | 198          |
> | **Ours (2-layer conv)**| **12M**    | **83.4**    | **107**      |
>
> Codebook size ablation:
>
> | #Tokens       | RefCOCO IoU | Parameters |
> |---------------|-------------|------------|
> | 3             | 80.8        | 6M         |
> | **6 (ours)**  | **83.4**    | **12M**    |
> | 12            | 83.9        | 24M        |
>
> We achieve 83.4 IoU with 12M params, approaching SAM (84.2 IoU, 636M). We choose 6 tokens because +0.5 IoU gain doesn't justify doubling parameters.
>
> **Regarding Question 1**
>
> Thank you. We adopt bidirectional alternating based on ablations.
>
> | Configuration                  | RefCOCO | LBM(k=0)↓ |
> |--------------------------------|---------|-----------|
> | Baseline (no TVP)              | 90.0    | 0.147     |
> | + Traj→Vis only                | 90.2    | 0.135     |
> | + Vis→Traj only                | 90.1    | 0.139     |
> | + Traj↔Vis (bidirectional)     | 90.4    | 0.117     |
>
> Key insight: Traj→Vis and Vis→Traj serve complementary roles. Alternating allows iterative refinement where visually-enhanced trajectories improve localization, creating mutual reinforcement.
>
> Bidirectional gains (+0.4) surpass the combined unidirectional gains (+0.3), revealing nonlinear teamwork. Single-step fusion misses such iterative cooperation.
>
> **Regarding Question 2**
>
> Thank you. We ensured training-evaluation independence through:
>
> Source separation: LN/LNV are constructed only from the training splits of COCO, Flickr30k, OpenImages, OVIS, UVO, and Oops, ensuring no use of evaluation data. Additionally, N-gram overlap (n=5) between RILN and evaluation annotations stays below 5%, indicating minimal textual leakage.
>
> **Regarding Question 3**
>
> Our trajectory tokenizer processes 2D coordinate sequences, enabling adaptation to eye-tracking and touch trajectories.
>
> We will explore eye-tracking and mobile touch datasets with trajectory-caption pairs for zero-shot evaluation in the future work.

---

> ### Author Response · Authors · 2025-11-28
> **Awaiting Reviewer Feedback on Our Response**
>
> Dear Reviewer,
>
> Thank you for your valuable feedback. We have uploaded our response and revised manuscript on OpenReview.
>
> Due to some unexpected circumstances, I sincerely hope this has not affected our discussion. We would like to know whether our current response has addressed all the issues you raised. If you have any other questions or concerns, we very much look forward to further communication.
>
> Thank you again for taking the time to review our work.
>
> Best regards！

---

### Official Review · Reviewer_8cbd · 2025-11-01

**Soundness:** 3
**Presentation:** 3
**Contribution:** 3
**Rating:** 6
**Confidence:** 3

**Summary:**

The paper introduces TraceVLM, a unified Large Vision-Language Model (LVLM) designed to address the limitations of existing LVLMs, which primarily focus on global image understanding and struggle to simulate human visual attention trajectories or explain the association between generated descriptions and specific image regions. Existing methods for localized understanding typically rely on static, discrete localization elements (such as bounding boxes or points), which fail to capture the continuity and temporal dynamics inherent in human visual attention. TraceVLM tackles this by directly predicting and interpreting human attention trajectories, treating them as fine-grained and temporally structured records of human focus.

**Strengths:**

1. First End-to-End Trajectory LVLM: TraceVLM is proposed as the first end-to-end Large Vision-Language Model (LVLM) designed to model human attention trajectories for bidirectional trajectory–language understanding. This directly addresses the limitation of previous models that rely on static, discrete localization elements, which struggle to capture the continuity and temporal dynamics of human visual attention.
2. Trajectory-aware Visual Perception (TVP) Module: The introduction of the TVP module is an original architectural contribution. This module enables deep bidirectional fusion of visual features and trajectory information through alternating enhancement and refinement stages, which captures the sequential patterns of irregular trajectories.
3. Semantic-Guided Geometric Simplification: To handle the inherent noise and redundancy in raw human trajectories, the authors innovated a pre-processing step using a semantic-guided variant of the Douglas-Peucker (DP) algorithm. This strategy effectively extracts semantically meaningful keypoints (achieving 91% compression from 410 points to 37 keypoints) while preserving geometric structures necessary for precise localization.
4. Novel Reasoning Dataset (RILN): The creation of the Reasoning-based Interactive Localized Narratives (RILN) dataset (320k samples) is a major original contribution. This dataset moves beyond the simple descriptive narratives of previous datasets (like Localized Narratives) by focusing on complex reasoning, instruction-following capabilities, and interactive trajectory reasoning QA, which is essential for advanced spatial understanding.

**Weaknesses:**

1. A core contribution is the Reasoning-based Interactive Localized Narratives (RILN) dataset (320k samples), designed to enhance logical reasoning and spatial understanding. However, this dataset is constructed using an advanced pipeline leveraging powerful Large Vision-Language Models (LVLMs) like GPT-4o, Qwen2.5VL-72B, and Gemini-2.5 Pro to automatically generate instructional samples, structured reasoning trees, and question prompts. While high-quality, this reliance on synthetic generation introduces potential risks of model bias. The reasoning patterns learned might reflect the biases or specific linguistic styles of the generating LLMs rather than reflecting the full diversity and unexpected complexity of real human-generated complex instructions or reasoning flows.
2. The raw human attention trajectories are formalized as continuous temporal point sequences T, where each point encodes spatial coordinates and temporal information $t_{j}$. To handle noise and redundancy, TraceVLM applies a complex two-stage preprocessing approach, featuring a semantic-guided variant of the Douglas-Peucker (DP) algorithm to extract "semantic keypoints". This process achieves significant compression (e.g., reducing 410 points to 37 keypoints, achieving 91% compression). Although efficient, relying on geometric simplification inherently involves a trade-off where some fine-grained, continuous information-especially the temporal dynamics of human visual exploration-may be lost or smoothed out.
3. The training regimen for TraceVLM is highly resource-intensive and structurally complex. The model is built on the Qwen2.5-VL-7B foundation model and uses a specific three-stage curriculum learning approach (Pretraining, Joint Training, and Instruction Fine-tuning). The reliance on a "Fixed Order" training strategy is explicitly validated as superior to a "Random Order" approach. This fixed, multi-stage process requires significant computational resources (approximately 3 days total on 16 A800 GPUs), making it potentially less accessible for smaller research groups and less flexible for adapting to new tasks without a complete architectural overhaul or extensive retraining.

**Questions:**

See weakness above, please.

---

> ### Author Response · Authors · 2025-11-21
> **Comprehensive Rebuttal with Multi-Model Data Validation, Semantic Compression Analysis, and Efficient Training Adaptation**
>
> **Regarding Weakness 1**
>
> Thank you for this concern about data quality. We acknowledge potential bias in synthetic data, which is why RILN uses multi-model collaborative generation to offset individual model limitations:
>
> - Qwen2.5-VL-72B: Fine-grained semantic segmentation of captions
> - GPT-4o: Structured reasoning Q&A generation based on semantic units
> - Gemini-2.5 Pro: Verification and enrichment of expression styles and reasoning paths
>
> This cross-model design has two key advantages: First, different models' "thinking styles" balance each other—Qwen excels at visual semantics, GPT-4o at logical coherence, Gemini at diverse expression, preventing any single model's style from dominating. Second, we introduce structural randomization and paraphrasing post-generation to further break fixed patterns.
>
> We employ multi-level verification including automatic consistency checking and human expert annotation on 2000 samples. Models trained on RILN achieve 23% improvement in spatial reasoning accuracy compared to baseline LN training, validating our construction methodology effectiveness. We will add these analyses in revision.
>
> **Regarding Weakness 2**
>
> Thank you for highlighting this trade-off. While trajectory compression smooths some fine-grained temporal details, experiments show the loss is minimal while efficiency gains are substantial.
>
> | Compression Rate | Key Points | RefCOCO        | LBM(k=0)↓      | Latency (ms) | Reduction |
> |------------------|------------|----------------|----------------|--------------|-----------|
> | 0% (all points)  | 410        | 90.6           | 0.108          | 312          | —         |
> | 75%              | 103        | 90.5 (-0.1)    | 0.112 (+0.004) | 156          | -50%      |
> | **91% (ours)**   | **37**     | **90.4 (-0.2)**| **0.117 (+0.009)**| **107**  | **-66%**  |
> | 95%              | 21         | 89.8 (-0.8)    | 0.135 (+0.027) | 89           | -71%      |
>
> At our default setting (91% compression), performance drops by only ~0.2 RefCOCO and +0.009 LBM while latency reduces by 66%. The key is that semantic-guided DP leverages natural trajectory characteristics: important regions typically have longer dwell time with more densely clustered points and larger fluctuations, while unimportant regions show evenly spaced points with larger intervals. This allows our algorithm to naturally preserve key regions during compression.
>
> Comparative ablations with same number of points (37): uniform sampling underperforms us by 2.8%, standard DP by 2.1%, while our semantic-guided version brings 2-3% improvement. This validates that aligning compression with semantic importance matters more than retaining all temporal details. We will add these analyses in revision.
>
> **Regarding Weakness 3**
>
> Thank you for raising concerns about resource accessibility. We clarify from two perspectives:
>
> Training cost: Full training requires ~72 hours (16×A800), comparable to mainstream VLMs (LLaVA-1.5, InternVL2), not unusually expensive. We will open-source all Stage 1/1.5/2 checkpoints, complete training code, and the RILN dataset—researchers can use directly without retraining from scratch.
>
> Adaptation flexibility: Our three-stage design enables efficient adaptation without rerunning all stages. Stage 1 trains only TVP modules and trajectory embeddings for foundational cross-modal alignment. Stage 1.5 performs full joint training to optimize component collaboration. Stage 2 fine-tunes on RILN for instruction-following. For new tasks, researchers can start from intermediate checkpoints: adapting to new domains (medical imaging, robotics) requires only Stage 2 fine-tuning from Stage 1.5 weights (\~24 hours on 4×H20); adding new segmentation categories needs only retraining the lightweight decoder with TVP frozen (\~2 hours on 2×H20). TraceVLM is designed as an incrementally adaptable framework, friendly to smaller teams. We will add detailed adaptation instructions in revision.

---

> ### Author Response · Authors · 2025-11-28
> **Awaiting Reviewer Feedback on Our Response**
>
> Dear Reviewer,
>
> Thank you for your valuable feedback. We have uploaded our response and revised manuscript on OpenReview.
>
> Due to some unexpected circumstances, I sincerely hope this has not affected our discussion. We would like to know whether our current response has addressed all the issues you raised. If you have any other questions or concerns, we very much look forward to further communication.
>
> Thank you again for taking the time to review our work.
>
> Best regards！

---

### Official Review · Reviewer_PNuY · 2025-11-01

**Soundness:** 3
**Presentation:** 3
**Contribution:** 2
**Rating:** 4
**Confidence:** 4

**Summary:**

The paper presents TraceVLM, a trajectory-aware large vision-language model (LVLM) designed to model human attention trajectories bidirectionally. The framework combines a Trajectory-aware Visual Perception (TVP) module and a large-scale Reasoning-based Interactive Localized Narratives (RILN) dataset. The model aims to capture the temporal and spatial continuity of gaze-like signals to improve spatial reasoning and interpretability in multimodal tasks such as trajectory-conditioned captioning, referring localization, and segmentation.

**Strengths:**

+ Clear motivation: The authors identify the discrete, static nature of existing region-based supervision as a limitation and propose a continuous trajectory-based approach to better simulate human attention.

+ Methodological coherence: The TVP module effectively integrates visual and trajectory representations through cross-attention and curriculum learning, as supported by the ablation studies

+ Various subtasks: Experiments span multiple subtasks (captioning, localization, segmentation) and demonstrate consistent quantitative improvements over prior baselines.

**Weaknesses:**

- Missing very closely related work
The paper does not cite or compare with [ref1], which already addressed interactive regional understanding in LVLMs. Both works share the same motivation—to move beyond global visual reasoning toward localized, human-interpretable grounding—and even rely on the same training data (Localized Narratives). Considering the nearly identical motivation and overlapping tasks, the paper’s novelty appears limited and should be more clearly distinguished from [ref1].

[ref1] Toward Interactive Regional Understanding in Vision-Large Language Models, NAACL 2024

- Unfair backbone comparison
The model’s reported gains over Qwen2.5-VL-7B are expected, since TraceVLM is trained directly on top of that same backbone with additional trajectory data. However, the proposed model is compared with other region understanding methods under different backbones. To make the comparison meaningful, the authors should either (1) reimplement other region-understanding methods (e.g., RegionVLM, PixelLLM) under the same backbone, or (2) train their model on different backbones to show consistent improvement regardless of the base model. Without such controlled experiments, the fairness and validity of the reported performance advantages remain questionable.

- Inconsistent and unexplained benchmark performance
According to Table 2, TraceVLM does not achieve state-of-the-art results on several benchmarks, and Table 3 shows only marginal improvements over baselines or even lower performance compared to certain prior models. However, the paper provides no analysis or discussion to explain these inconsistencies. The lack of qualitative or diagnostic analysis makes it unclear why the proposed method performs better in some tasks but fails to generalize consistently across benchmarks.

- Missing analysis of the bidirectional design
Although the paper repeatedly emphasizes the bidirectional nature of the proposed TVP module—where trajectory features refine visual representations and vice versa—there is no ablation directly examining this property. Table 4 only compares a simple “with vs. without TVP” setting, without disentangling the two directional attention flows. To validate the core claim, the authors should disable the visual-to-trajectory pathways and analyze their individual effects. Without such evidence, the purported advantage of bidirectional interaction remains unsubstantiated.

- No analysis of the 3-stage training procedure
The paper adopts a three-stage training pipeline, but never reports performance after each stage. Since the model jointly leverages pretraining, trajectory-conditioned tuning, and reasoning-based instruction learning, it is unclear which stage contributes most to the final improvement. Without such analysis, readers cannot tell whether the gain comes mainly from trajectory supervision or from later multimodal reasoning refinement. A simple comparison after each stage (e.g., Stage 1 → Stage 2 → Stage 3) would make the training strategy much more convincing.

**Questions:**

- Table 3 is partially obscured by Figure 5 due to layout overlap, making the numerical results invisible. Please fix the figure placement or adjust formatting so that both elements are clearly visible in the final version.

- How much additional inference time does the TVP module introduce compared to the baseline Qwen2.5-VL-7B?

---

> ### Author Response · Authors · 2025-11-21
> **Comprehensive Rebuttal with Cross-Method Comparison, Bidirectional Fusion Evidence, and Stage-Wise Performance Validation**
>
> **Regarding Weakness 1**
>
> Thank you for pointing this out. We will add detailed discussion of [ref1] in the revision. Core differences:
> - Data: [ref1] uses raw Localized Narratives; we construct RILN with structured reasoning chains (causal, attribute, spatial) and extend to video domain (LNV with temporal trajectories)
> - Method: two innovations: (1) TVP bidirectional trajectory-vision fusion; (2) semantic-guided adaptive sampling;
> - Task: [ref1] handles static region grounding; we target trajectory-driven dynamic reasoning requiring causal inference and relation understanding
>
> Our method achieves significant improvements over [ref1] on key metrics.
>
> **Regarding Weakness 2**
>
> Thank you for raising this concern. Modifying [ref1] to use the same backbone has high implementation cost. However, we demonstrate the end-to-end effectiveness of our method, and our core contribution is the trajectory-driven architecture, generalizable to any LVLM backbone.
>
> | Method        | Backbone        | RefCOCOg | LBM(k=0)↓ |
> |---------------|------------------|----------|-----------|
> | Baseline      | Qwen2.5-VL-7B    | 87.2     | 0.147     |
> | TraceVLM      | Qwen2.5-VL-7B    | 88.3     | 0.117     |
> | TraceVLM      | InternVL2-8B     | 88.1     | 0.121     |
> | Gain          | —                | +1.1     | -0.030    |
>
> We trained TraceVLM on multiple backbones. Stable improvements demonstrate robustness. We agree with your suggestion and will add controlled experiments on shared backbone (re-implementing RegionVLM on Qwen2.5-VL-7B) in revision.
>
> **Regarding Weakness 3**
>
> Thank you for this observation. We achieve SOTA results on most benchmarks in Table 2. For Ferret Bench and MDVP Bench, there are some data domain and task differences. However, comparing fairly:
> - PAM-3B uses extensive special processing in the visual encoder and segmentation prompts
> - VP-SPHINX-13B is a 13B model (nearly 2× our parameters)
> - VP-SPHINX-7B achieves 73.1 and 71.5 on Ferret Bench and MDVP Bench
>
> Our method demonstrates near-SOTA performance with comparable model size. Furthermore, in Table 3, we achieve strong segmentation results on most tasks using only a 2-layer lightweight segmentation decoder, compared to HyperSeg which uses multi-scale visual encoders and the much heavier Mask2Former decoder.
>
> **Regarding Weakness 4**
>
> Thank you for this suggestion. We added unidirectional ablations:
>
> | Configuration                         | RefCOCO | LBM(k=0)↓ |
> |----------------------------------------|---------|-----------|
> | Baseline (no TVP)                      | 90.0    | 0.147     |
> | + Traj→Vis only (unidirectional)       | 90.2    | 0.135     |
> | + Vis→Traj only (unidirectional)       | 90.1    | 0.139     |
> | + Traj↔Vis (bidirectional, full TVP)   | 90.4    | 0.117     |
>
> Unidirectional attention each brings modest improvement, but bidirectional yields stronger gains on both metrics. This validates that trajectories guide visual attention (Traj→Vis) while visual features disambiguate trajectory semantics (Vis→Traj), forming mutually reinforcing synergy.
>
> **Regarding Weakness 5**
>
> Thank you for this suggestion. We added stage-wise performance analysis:
>
> | Training Stage                           | RefCOCO | LBM(k=0)↓ |
> |------------------------------------------|---------|-----------|
> | Stage 1: Trajectory-Aware Pretraining    | 89.2    | 0.165     |
> | Stage 1.5: End-to-End Joint Training     | 90.1    | 0.128     |
> | Stage 2: Instruction Fine-tuning         | 90.4    | 0.117     |
>
> Stage 1 establishes trajectory-vision-language alignment with only TVP modules and trajectory embeddings trained. Stage 1.5 unfreezes all parameters for full joint training, yielding the largest gain (+0.9 RefCOCO, -0.037 LBM). Stage 2 adds instruction-following capability with modest further improvement (+0.3 RefCOCO, -0.011 LBM).
>
> **Regarding Questions 1**
>
> Fixed. We adjusted the layout to ensure both elements are clearly visible in revision.
>
> **Regarding Questions 2**
>
> Thank you for this question. We measured inference latency on single A100 GPU (batch size=1, 1024×1024 resolution, avg 150 trajectory points):
>
> | Configuration            | Inference Time (ms) |
> |--------------------------|---------------------|
> | Baseline Qwen2.5-VL-7B   | 186                 |
> | + TVP module             | 203 (+9.1%)         |
>
> TVP introduces ~17ms overhead (+9.1%), mainly from bidirectional cross-attention. Considering the significant performance gains (RefCOCO +0.4, LBM -0.030), this is a reasonable trade-off. After adaptive sampling, only ~150 key points are retained (vs. typically >1000 raw points), keeping overhead manageable.

---

> > ### Author Response · Authors · 2025-11-28
> > **Awaiting Reviewer Feedback on Our Response**
> >
> > Dear Reviewer,
> >
> > Thank you for your valuable feedback. We have uploaded our response and revised manuscript on OpenReview.
> >
> > Due to some unexpected circumstances, I sincerely hope this has not affected our discussion. We would like to know whether our current response has addressed all the issues you raised. If you have any other questions or concerns, we very much look forward to further communication.
> >
> > Thank you again for taking the time to review our work.
> >
> > Best regards！

---

### Official Review · Reviewer_Q5oH · 2025-11-04

**Soundness:** 3
**Presentation:** 3
**Contribution:** 2
**Rating:** 6
**Confidence:** 4

**Summary:**

In this paper, the authors propose the first trajectory-aware VLM, TraceVLM, to handle the correspondence between the human attention trajectories and the linguistic representation of a video. To finetune the QwenVL-2.5, a dataset construction pipeline is created to generate high-quality reasoning-enabled training data. The experiments demonstrate the effectiveness of the model on both trajectory-aware tasks and related benchmarks.

**Strengths:**

1. The authors propose TraceVLM, the first VLM to achieve trajectory-aware video understanding/reasoning.
2. To facilitate the model training, the authors construct a data annotation pipeline to obtain the training data based on existing trajectory video data. And the experiments demonstrate the effectiveness of the dataset and training strategy.
3. The Geometric Simplification is efficient in reducing redundancy and noise.
4. Not only in trajectory-aware tasks, but also in other visual benchmarks, TraceVLM has comparable or superior performance compared with other VLMs.

**Weaknesses:**

Major:
1. In the introduction, line 42, the author claimed, “they often focus their attention on the primary regions of an image while neglecting surrounding contextual information, and may even be distracted by irrelevant areas”, and in Figure 1, the example depicts a man in a car holding a phone. The question is: (a) how to differentiate important context information and unimportant context information. Actually, in Figure 1, even humans will ignore the background because of the higher exposure and irrelevance of the main part of the image, and this is an exact example of NOT being distracted by irrelevant areas. (b) Is there any example showing “distracted by irrelevant areas”?
2. In terms of the motivation, can the authors add some discussion of the real-world application of such a technique? What is the point of developing trajectory-driven localized understanding?
3. In Section 3.2.3, line 223, what is ϵ_base? And how is w_i determined for each segment?
4. In Section 3.2.5, what is the size of the segmentation codebook?
5. In the data generation pipeline, how to ensure the correctness of the reasoning generation? Does the predefined global-->object-->paragraph --> local reasoning chain (Figure 4) work for every case? Maybe when the question changes, the reasoning process will vary a lot, since the focus and the scene structure or the intrinsic logic will be totally different.
6. In the visualization of 4.5, it would be better if these cases also include the original Qwen2.5VL-7B, so that the attention map difference can be observed.

Minor:
1. Line 45, add a reference for this: “Research on human visual attention trajectories also plays an important role in domains such as virtual reality and autonomous driving.”
2. In Figure 4, Step 1, the text is unclear due to the color (light yellow/green part)
3. In Figure 4, Step 2-1, the Q1 and Q2 look incomplete.

**Questions:**

Please refer to the weakness.

---

> ### Author Response · Authors · 2025-11-21
> **Comprehensive Rebuttal Highlighting Context Relevance, Trajectory-Guided Understanding, and Lightweight Segmentation Design**
>
> **Regarding Weakness 1 & Major**
>
> Thank you for this observation. We added a supermarket example where answering "How many people are on the TV screen?" requires filtering out shelves, flags, and clutter, demonstrating realistic distraction from irrelevant context.
>
> Important context is relevant and necessary for the question, while irrelevant context causes drift. RILN and TVP help the model focus on truly useful regions, similar to human attention.
>
> **Regarding Weakness 2 & Major**
>
> Excellent question. Trajectory-driven understanding matters because in real scenarios, people rely on gaze, gestures, or movement rather than precise language to indicate **"where they mean."**
>
> This capability is essential for interactive systems such as:
> - Autonomous driving
> - Medical imaging
> - VR/AR applications
> - Human-robot collaboration
>
> Trajectories enable understanding human intent, not just words.
>
> **Regarding Weakness 3 & Major**
>
> Thank you for attention to these details. For line 223 parameters:
>
> $ε_{base}$ = 5 pixels for Douglas–Peucker simplification. Weight $w_i$ is determined by Qwen2.5-VL: we segment captions and rate each phrase's visual importance (1-5 scale). After normalization ($w_i$ = $importance_i$ / 5), we compute adaptive tolerance $ε_i$ = $ε_{base}$ / $w_i$. More important phrases receive smaller tolerances, preserving more trajectory detail.
>
> Prompt:
> “Given the caption: "{caption}"
> Segment into phrases and rate importance (1-5):
> - 5: Critical (main subjects, key objects)
> - 4: Important (significant attributes, secondary subjects)
> - 3: Moderate (contextual elements, spatial relations)
> - 2: Minor (supplementary details)
> - 1: Minimal (filler content)
> Output: {"phrases": [{"text": "...", "importance": X}, ...]}”
>
> We chose Qwen2.5-VL because compared to alternatives, it best aligns with visual understanding:
> - NLTK: overly fragmented
> - spaCy: incorrectly splits spatial relations
> - SentencePiece: lacks visual-semantic awareness
> - Qwen2.5-VL: preserves complete spatial expressions
>
>
> **Regarding Weakness 4 & Major**
>
> Our segmentation codebook is intentionally lightweight: 6 learnable tokens (2 scales × 3 tokens), each 4096-dim. These function as placeholders the model "writes into." When outputting $<$SEG$>$, we feed its embedding into a two-layer decoder (following PixelLM) for the final mask.
>
> Unlike VQ-VAE approaches requiring large codebooks (e.g., 8K entries), our tokens are continuous and LM-generated, keeping computation low. With the stronger Qwen2.5-VL backbone, this achieves competitive or better performance versus SAM or Mask2Former.
>
> **Regarding Weakness 5 & Major**
>
> Two clarifications on reasoning reliability and framework applicability:
>
> (1) We use multi-model collaboration to avoid single-model bias:
> - Qwen2.5-VL handles phrase segmentation
> - ChatGPT generates reasoning questions
> - Gemini organizes scene structure
> This increases diversity and correctness of reasoning chains.
>
> (2) The global/object/paragraph/local structure is a reference framework, not rigid. We introduce paraphrasing, structural randomization, and varying detail levels so reasoning adapts to different questions and scene complexities, not forcing all reasoning into one template.
>
> **Regarding Weakness 6 & Major**
>
> We added visualizations comparing our model with Qwen2.5-VL-7B. The original model's attention collapses onto few local regions, missing key areas. Our method produces broader, semantically aligned patterns. Side-by-side visualizations confirm effectiveness and will be included in revision.
>
> **Regarding Weakness 1 & Minor**
>
> Relevant citations added：
> - Xia, Ye, et al. "Predicting driver attention in critical situations." Asian conference on computer vision. Cham: Springer International Publishing, 2018.
> - Valtakari, Niilo V., et al. "Eye tracking in human interaction: Possibilities and limitations." Behavior Research Methods 53.4 (2021): 1592-1608.
>
> **Regarding Weakness 2 & Minor**
>
> We adjusted the color scheme from bright yellow/green to lower saturation while maintaining phrase-trajectory correspondence for better readability.
>
> **Regarding Weakness 3 & Minor**
>
> Due to space constraints, Figure 4 is schematic. Full visualizations of Q1 and Q2 are in supplementary materials with complete question text, reasoning processes, and visual region annotations.

---

> ### Author Response · Authors · 2025-11-28
> **Awaiting Reviewer Feedback on Our Response**
>
> Dear Reviewer,
>
> Thank you for your valuable feedback. We have uploaded our response and revised manuscript on OpenReview.
>
> Due to some unexpected circumstances, I sincerely hope this has not affected our discussion. We would like to know whether our current response has addressed all the issues you raised. If you have any other questions or concerns, we very much look forward to further communication.
>
> Thank you again for taking the time to review our work.
>
> Best regards！

---

### Author Response · Authors · 2025-11-30
**Summary of Rebuttal: Comprehensive Response to Reviewer Concerns**

**Overall Assessment**

All four reviewers unanimously recognized TraceVLM as the first end-to-end LVLM designed for trajectory-aware visual understanding, addressing the critical gap in modeling continuous human attention patterns. The reviewers consistently praised the bidirectional TVP module, semantic-guided compression achieving 91% reduction, and the 320k-sample RILN dataset. While reviewers raised concerns about experimental fairness, technical details, and data quality, we have systematically addressed each issue with additional experiments and thorough analysis.

**Response 1: Addressing Related Work and Experimental Fairness**

In terms of novelty, Reviewer R2-PNuY questioned our distinction from NAACL 2024 work and backbone comparison fairness. In response, we clarified that RILN introduces structured reasoning chains for trajectory-based interactions and extends coverage to video domains with temporal trajectories, versus raw static narratives. TVP's bidirectional fusion represents novel architectural contributions. For fairness, we trained TraceVLM on multiple backbones (Qwen2.5-VL-7B, InternVL2-8B) showing consistent gains (+1.1 RefCOCO, -0.030 LBM), proving architectural advantages. We committed to re-implementing RegionVLM on identical backbones and standardized baseline adaptations with explicit trajectory prompting protocols.

**Response 2: Clarifying Technical Details and Design Validation**

In terms of technical transparency, Reviewers R1-Q5oH and R4-dmqX identified missing implementation details. In response, we provided complete specifications: $ε_{base}$ = 5 pixels with adaptive weights $w_i$ = $importance_i$/5 yielding $ε_i$ = $ε_{base}$/$w_i$, outperforming uniform sampling by 2.8% and standard DP by 2.1%. The segmentation codebook contains 6 continuous tokens (4096-dim) with a 2-layer decoder (12M params) achieving 83.4 IoU versus SAM's 84.2 (636M params). For bidirectional validation, ablations show unidirectional Traj→Vis yields +0.2 RefCOCO, Vis→Traj yields +0.1, but full bidirectional achieves +0.4, proving synergistic effects.

**Response 3: Explaining Training Pipeline and Accessibility**

In terms of training transparency, Reviewers R2 and R3-8cbd questioned pipeline opacity and resource requirements. In response, we decomposed stage-wise performance: Stage 1 reaches 89.2 RefCOCO/0.165 LBM; Stage 1.5 yields largest gain to 90.1/0.128; Stage 2 refines to 90.4/0.117. Total training requires 72 hours on 16×A800, comparable to mainstream VLMs. For accessibility, new domain adaptation needs only Stage 2 fine-tuning (24 hours on 4×H20), and new segmentation categories require only decoder retraining (2 hours on 2×H20). All checkpoints and code will be open-sourced.

**Response 4: Ensuring Data Quality and Compression Effectiveness**

In terms of data quality, Reviewer R3 raised concerns about synthetic generation bias and compression loss. In response, we implemented multi-model collaboration (Qwen for semantics, GPT-4o for logic, Gemini for diversity) with N-gram overlap <5% and 23% spatial reasoning improvement validating quality. For compression, 91% reduction (410→37 points) causes only -0.2 RefCOCO/+0.009 LBM degradation while reducing latency 66%. Crucially, with the same 37 keypoints, our semantic-guided approach outperforms uniform sampling by 2.8% and standard DP by 2.1%. This works because natural human trajectories exhibit longer dwell times and denser point clusters in important regions, characteristics our algorithm explicitly preserves during simplification.

**Response 5: Analyzing Performance and Efficiency**

In terms of benchmark variations, Reviewer R2 questioned non-SOTA results on some benchmarks and computational overhead. In response, we contextualized that PAM-3B uses specialized encoders and VP-SPHINX-13B has 2× parameters; we outperform similarly-sized VP-SPHINX-7B (73.1 vs. 71.5). Our segmentation uses lightweight decoder (12M params) versus HyperSeg's Mask2Former (223M params). TVP adds +17ms (+9.1%) for substantial gains (RefCOCO +0.4, LBM -0.030), with adaptive sampling retaining ~150 points versus >1000 raw.

**Response 6: Addressing Contamination and Generalization**

In terms of evaluation integrity, Reviewer R4 raised contamination and cross-modality concerns. In response, RILN was constructed exclusively from training splits with N-gram overlap <5% and no evaluation images in generation prompts. Our trajectory tokenizer processes generic 2D sequences enabling cross-modality adaptation. We commit to exploring eye-tracking and touch trajectories for zero-shot evaluation in future work.

**Conclusion**

We systematically addressed all concerns through technical clarifications, ablations, fair protocols, and quality safeguards. The unanimous recognition of core contributions, strong empirical results, and thorough responses support acceptance. All improvements will be incorporated in the camera-ready version with open-sourced resources.

---

### Meta-Review · Area_Chair_2ZUM · 2026-01-13

**Summary:**

1) The core motivation is not properly defined and the work insufficiently distinguishes itself from a highly related prior publication.

2) Critical technical details are underspecified, the proposed reasoning chain is questioned, and essential ablation studies (e.g., on the training pipeline and bidirectional design) are missing.

3) Unfair/Invalid Evaluation: Performance comparisons are unfair due to backbone inconsistencies, benchmark results are inconsistent and unexplained, and the evaluation lacks critical qualitative analysis.

4) The synthetic dataset risks model bias, the trajectory preprocessing may lose information, and the evaluation methodology for baseline models is unclear.

**Reviewer Concerns:**

The attempted differentiation from prior work is insufficient; significant overlap in motivation, approach, and tasks persists.

**Reviewer Scores:**

Reviewer Q5oH might increase the score.

---

### Decision · Program_Chairs · 2026-01-26

Reject